# HarnessLLM: Automatic Testing Harness Generation via Reinforcement Learning

## Abstract

Existing LLM-based automatic test generation methods mainly produce input and expected output pairs to categorize the intended behavior of correct programs. Although straightforward, these methods have limited diversity in generated tests and cannot provide enough debugging information. We propose HarnessLLM, a two-stage training pipeline that enables LLMs to write harness code for testing. Particularly, LLMs generate code that synthesizes inputs and validates the observed outputs, allowing complex test cases and flexible output validation such as invariant checking. To achieve this, we train LLMs with SFT followed by RLVR with a customized reward design. Experiments show that HarnessLLM outperforms input-output-based testing in bug finding and testing strategy diversity. HarnessLLM further benefits the code generation performance through test-time scaling with our generated test cases as inference-phase validation.

## 1 Introduction

Large language models (LLMs) have demonstrated remarkable proficiency in code-related tasks, including code generation, completion, and even resolving software engineering issues through tool use Chen et al. (2021); Li et al. (2022); OpenAI (2024); DeepSeek-AI (2025); Jimenez et al. (2024); Li et al. (2025). However, compared to these code generation tasks, automatic testing and debugging AI-generated programs have received comparatively little attention, even though comprehensive test suites are critical for ensuring the correctness and robustness of the AI-generated code Chen et al. (2024a); Prasad et al. (2025); Sinha et al. (2025); He et al. (2025); Zhang et al. (2023a).

Existing works in automatic testing mainly prompt the language model to generate input–output pairs that characterize the intended behavior of correct programs Chen et al. (2022); Prasad et al. (2025); Zeng et al. (2025); Lin et al. (2025). As depicted in Figure 1, the model produces examples of test inputs alongside their expected test outputs. The target program is then executed on a test input, and the output that the program generates is compared against the corresponding expected test output. A bug is exposed if the two outputs diverge.

Although straightforward, such an input-output test case generation paradigm has two potential drawbacks. *First*, the test inputs generated by the language model tend to be simple and homogeneous, so they may not have sufficient coverage of the sophisticated corner cases that could expose bugs. *Second*, such a paradigm requires that the model generates the correct output by itself, which becomes extremely challenging for complicated programming tasks or for complicated test inputs. In short, the fundamental paradox is that the 'tester', *i.e.,* the language model that generates test cases, is often much weaker than the 'testee', *i.e.,* the program, in accomplishing the complex programming tasks (otherwise, the language models would not have to rely on code generation to solve these tasks).

In this paper, we explore a novel debugging paradigm that could resolve this paradox – LLM-based *test harness generation*. Rather than letting the language model directly generate input-output pairs, we prompt it to write executable code, a matching rival to the testee, to generate test inputs and validate target program outputs. In this way, both aforementioned drawbacks can be addressed simultaneously. On the input side, executable code can easily generate various richly structured, diverse, and complicated inputs. On the output side, executable code opens up many possibilities to validate target program outputs. It can ❶ directly generate hardcoded expected output, as does the input-output paradigm, or ❷ write a reference program to compute the expected output, or, most interestingly ❸ assert output properties and requirements. As shown in Figure 1, for a program that

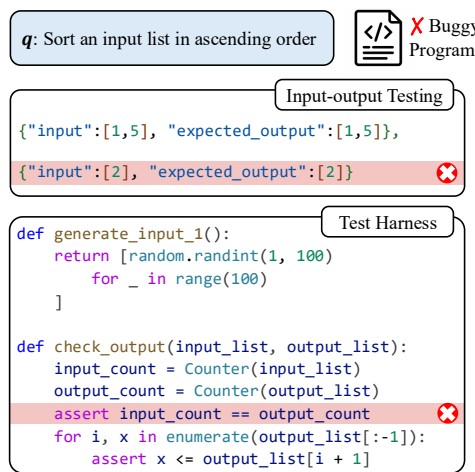

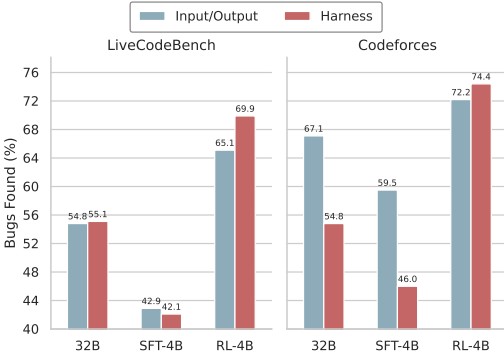

Figure 2: Percentage of found bugs (average of 8 runs, higher is better) for two strategies with different models.

Figure 1: Comparison between input-output pairs (top) and test harness (bottom).

sorts a list of input integers, the LLM first writes an input generator, `generate_input_1`, to generate random lists, which are fed to the target program for execution. The returned outputs are then validated by an LLM-defined function, `check_output`, which checks that the result is sorted and preserves the original integers. With programmatic input generation and output validation, testing harnesses can support complex invariant checking and stress testing, enabling more comprehensive testing and detection of deep logical bugs.

However, the key challenge of the test harness generation paradigm is that even the strong coding LLMs are not inherently capable of test code generation, which requires different skills than generating code for programming tasks: The former requires understanding the given program's logic, control and data flow, designing proper stress tests, and writing validation logic, while the latter is mainly about writing code to fulfill the required functionality. To demonstrate this, our initial experiment compares the bug-finding rates of the input–output strategy versus test harness generation on the LIVECODEBENCH and CODEFORCES datasets Jain et al. (2025); Penedo et al. (2025), using a strong reasoning model Qwen3-32B Yang et al. (2025). Surprisingly, direct prompting for test harnesses does not yield better bug finding capabilities (Figure 2).

To close this gap, we propose HarnessLLM, a two-stage training pipeline combining supervised fine-tuning (SFT) with reinforcement learning (RL) with customized reward functions. First, we collect SFT data by prompting Qwen3-32B and filtering for harnesses that successfully expose a bug. We warm up a smaller model (*e.g.,* Qwen3-4B) with SFT on collected data. The purpose of this stage is to provide a reasonable starting point for reinforcement learning, which improves RL's training efficiency. Second, we further train the SFT model using RL with our customized verifiable outcome reward. Here, we assume access to a ground-truth program during training. To encourage the model to generate valid harnesses, we first give a zero reward to generated tests that trigger compilation or runtime errors on the ground-truth program. Then, we design rewards to incentivize the model to generate effective tests that crash the target programs. Specifically, a positive reward is assigned when the ground-truth program can pass the generated tests but the target program fails, indicating that the test harness correctly identifies bugs in the target program. We train the model to maximize the expected reward using the GRPO algorithm Shao et al. (2024). The RL training can further strengthen the model's capabilities to generate effective test harnesses and improve the model's generalizability.

We train on two base models (Qwen3-4B and Llama3.2-3B Llama (2024)) and evaluate on three benchmarks containing buggy programs. Experiments show that our model outperforms all baselines, including the off-the-shelf Qwen3-32B and another model that is also trained with RL but only generates input-output pairs (Figure 2 presents an overview). Moreover, the learned harness generator generalizes to code produced by unseen models and can be used for improving code generation performance. Specifically, using the execution results of generated test cases to select the best out of 8 responses improves Qwen3-32B's performance from 63.5% to 69.5% on LIVECODEBENCH

Jain et al. (2025). To the best of our knowledge, HarnessLLM is *the first LLM-based testing harness generation that enables comprehensive testing and benefits competitive programming tasks*.

We summarize our contributions as follows:

- We propose harness-based automatic program testing, a new debugging paradigm with richer context and more diverse testing cases beyond input-output checks.
- We design a pipeline with SFT and RL to train LLMs to write effective test harnesses.
- We trained specialized reasoning models using HarnessLLM, comparing their effectiveness with SOTA LLMs, and demonstrating their utility in code generation.

## 2 RELATED WORKS

**Automatic Test Case Synthesis.** Test cases are crucial in evaluating code correctness. While many established benchmarks rely on manually written test cases Chen et al. (2021); Austin et al. (2021); Hendrycks et al. (2021), this process is labor-intensive and does not scale well. To address this limitation, a variety of automatic test case synthesis methods have been proposed. Traditional approaches leverage programming language techniques to explore the input space and cover diverse execution paths Puspitasari et al. (2023); Forgács & Kovács (2024); Guo et al. (2024); Reid (1997). Although these techniques improve input coverage, they often fall short in capturing code semantic relationships and complex control flows, which can lead to undetected failures during runtime. Recently, LLMs have been used to synthesize test cases by prompting them to generate both inputs and expected outputs Yuan et al. (2024); Chen et al. (2024b); Han et al. (2024); Li & Yuan (2024); Guzu et al. (2025); Xiong et al. (2023); Wang et al. (2025a); Cao et al. (2025); Wang et al. (2025b). Despite their strong code understanding capabilities, LLMs still struggle to consistently generate correct outputs, especially when the code is complex. In this work, we propose a novel paradigm that shifts from output prediction to execution-based validation. Our HarnessLLM programmatically generates inputs and validates outputs, expanding the design space of test cases.

**Reinforcement Learning with Verifiable Rewards.** Reinforcement learning has shown great potential in improving LLM abilities in many domains requiring heavy reasoning, such as math problem solving DeepSeek-AI (2025); Kimi (2025); Shao et al. (2024); Yu et al. (2025); Hou et al. (2025), code generation Le et al. (2022); El-Kishky et al. (2025); Liu & Zhang (2025), and robotic control Chu et al. (2023); Ji et al. (2025). In this work, we use RL to improve LLMs' test case generation abilities. By designing a customized reward that judges whether the generated test cases can differentiate between correct and buggy programs, we train LLMs to learn the reasoning skills required to write effective test cases.

## 3 METHODOLOGY

### 3.1 PROBLEM FORMULATION

Formally, let $q$ be the description of a programming problem with input space $\mathcal{I}$ and output space $\mathcal{O}$. Denote $f, g : \mathcal{I} \to \mathcal{O}$ as two programs for this problem, where $f$ is a potentially buggy implementation that is under testing, and $g$ is a ground-truth implementation for the problem. We say $f$ has logical bugs if for some $\boldsymbol{x} \in \mathcal{I}, f(\boldsymbol{x}) \neq g(\boldsymbol{x})$. In other words, $\boldsymbol{x}$ triggers the divergent behaviors of the buggy and reference programs. Therefore, an automatic debugging method generally contains two steps: generating inputs that can potentially trigger the bug and comparing the target program's output with the reference output.

However, in most real-world situations, the ground-truth implementation $g$ is not available, which necessitates an approximate verifier to validate the output of $f$. Denote this verifier as $v : \mathcal{I} \times \mathcal{O} \to \{0, 1\}$, where $v(\boldsymbol{x}, \boldsymbol{y}) = 1$ indicates that output $\boldsymbol{y}$ on input $\boldsymbol{x}$ is deemed correct. Our goal in this paper is to train an LLM for automatic debugging that, given $q$ and $f$, emits both a set of inputs $\{\boldsymbol{x}_i\}_{i=1}^N$ and a corresponding verifier $v$. Note that we mainly focus on finding *logical bugs* in a target program, *i.e.,* deviations from intended behavior, and leave security vulnerability for future work.

**Challenge of Input-Output Testing.** The input-output testing can be considered as having a simple verifier that compares the program's output with the expected output. Specifically, the model

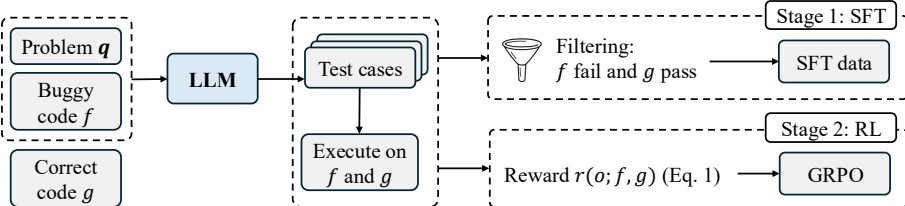

Figure 3: Overview of our training pipeline.

generates a set of pairs $\{(\boldsymbol{x}_i, \hat{\boldsymbol{y}}_i)\}_{i=1}^N$, where $\hat{\boldsymbol{y}}_i$ is the expected output for input $\boldsymbol{x}_i$. The verifier is then an indicator function $v(\boldsymbol{x}_i, f(\boldsymbol{x}_i)) = \mathbb{1}(f(\boldsymbol{x}_i) = \hat{\boldsymbol{y}}_i)$. However, this simple verifier requires the model itself to come up with a correct expected output, which limits the complexity of test cases. In the following, we propose a framework that generates test harnesses to address this challenge.

## 3.2 GENERATING TEST HARNESS FOR DEBUGGING

We propose instead that the LLM writes a test harness code that synthesizes inputs and programmatically checks outputs. Having harnesses can help produce more diverse testing cases and provide more valuable feedback when the program crashes. Specifically, our framework consists of three steps.

***Step 1: Generate Input.*** The model implements a set of input generators, each returning a list of inputs for the program (*e.g.,* `generate_input_1()`). By leveraging loops or random functions, the LLM can craft rich test inputs, which would be difficult to get with hardcoding.

***Step 2: Execute.*** Each generated input is fed to the program $f$, and the resulting output is captured.

***Step 3: Validate Output.*** A model-implemented function `check_output(input,output)` is used to validate the correctness of each captured output. The model can use various ways for validation, such as checking specific invariants or comparing with output from a brute-force implementation. This output checker uses assertions to check correctness, and a bug is reported if the assertions fail for *any* pair of generated input and captured output.

Figure 6 shows a complete example of model generation for this process, and Figure 11 shows the detailed prompt we use.

## 3.3 IMPROVING TEST HARNESS VIA RLVR

Despite the promise, we found off-the-shelf LLMs struggle to generate effective harnesses. To remedy this, we design a two-stage training pipeline to improve their performance. Figure 3 depicts an overview of our pipeline.

**Stage 1: SFT Warm-Up.** We prompt `Qwen3-32B` to generate test harnesses as described in Section 3.2. The model response contains a long reasoning chain and a final code block. We execute the harnesses against both the target program $f$ and the ground-truth program $g$ and retain only responses for which $g$ passes but $f$ fails. We then fine-tune a smaller model (*e.g.,* `Qwen3-4B`) with SFT on the filtered dataset. The SFT model has a basic understanding and skills for test harness generation. Using it as an initialization for RL can improve the learning efficiency of RL, as the early training stage can receive some meaningful positive rewards.

**Stage 2: RL with Verifiable Outcome Reward.** To further improve the generalizability of the warmed-up model, we follow recent works to train the model with RL against a verifiable outcome reward DeepSeek-AI (2025); Lambert et al. (2025). Specifically, for each rollout $o$ the model generates, let $\{\boldsymbol{x}_i\}_{i=1}^N$ be the corresponding inputs, we define the following reward function based on the execution results on $f$ and $g$:

$$r(o; f, g) = \begin{cases} 1, & \text{if } g \text{ passes and } f \text{ fails;} \\ 0.1, & \text{if } g \text{ fails}[1]\text{or } f \text{ passes, and } \exists \boldsymbol{x}_i : f(\boldsymbol{x}_i) \neq g(\boldsymbol{x}_i); \\ 0, & \text{otherwise} \end{cases} \quad (1)$$

In other words, a reward of 1 is given only when the ground-truth program can pass the test, but not the buggy program, indicating a correct test case. Otherwise, if all inputs are valid (*i.e.,* they do not trigger runtime errors on $g$) and at least one input can trigger different outputs for $f$ and $g$, we assign a partial reward of 0.1, which encourages the model to generate bug-exposing inputs. Note that in this case, the input generators work well, but the output verifier generates ineffective assertions, which either fail the correct code $g$ or do not crash the buggy code $f$. Nevertheless, we still assign a partial reward to incentivize the model to generate good inputs. Finally, a reward of 0 is given when no input can expose the bug. Importantly, the requirement that $g$ has to pass the generated test cases prevents the model from hacking rewards by generating arbitrary invalid tests. We maximize the expected reward using GRPO.

## 3.4 DATA COLLECTION

Both training stages in Section 3.3 require data in the format of a problem description $q$, a buggy implementation $f$, and a ground-truth implementation $g$. To collect such data, we follow prior works Luo et al. (2025) to source from existing datasets of coding problems, including TACO Li et al. (2023), SYNTHETIC-1 Intellect (2025), LeetCode Xia et al. (2025), and Codeforces MatrixStudio (2025). The original solution in the datasets is used as ground-truth program $g$, after an additional round of filtering to make sure $g$ passes all provided test cases of the problem.

To collect the buggy programs $f$, we prompt a series of LLMs to solve the problem, including `Qwen2.5-Coder` 1.5-7B Hui et al. (2024) and `DeepSeek-R1-Distill-Qwen-1.5B` DeepSeek-AI (2025). We only keep programs that satisfy both of the following conditions: ① The program passes the demo test cases in the problem description; and ② The program fails on at least one test case of the problem. This makes sure the retained programs are partially correct but still have bugs. We retain at most two buggy programs per problem and select the two that pass the most test cases if multiple programs satisfy the two conditions.

After decontamination against all evaluation data in Section 4.1, the resulting training set contains 12,043 unique $(q, f, g)$ triplets. We use all samples for RL training and a subset of 6,805 samples to generate SFT data. Appendix A.1 details the procedure for our data collection process.

## 4 EXPERIMENTS

We conduct experiments to validate the effectiveness of HarnessLLM. Specifically, we aim to answer two questions: ① Does our two-stage training pipeline enhance models' ability to write test harnesses? ② Does harness-based testing outperform input-output testing in identifying bugs?

## 4.1 EXPERIMENT SETTING

**Evaluation Benchmarks.** We evaluate on three widely used code generation datasets: MBPP+ Austin et al. (2021); Liu et al. (2023), LIVECODEBENCH Jain et al. (2025), and CODEFORCES Penedo et al. (2025). We repurpose these datasets for the bug detection task by collecting triplets of problem description, buggy program, and ground-truth program. For MBPP+, we directly use the split MBPP+FIX (HARD) in `UTGen-32B` Prasad et al. (2025). For LIVECODEBENCH and CODEFORCES, we follow the procedure described in Section 3.4. Particularly, we create two dataset variants: ❶ SEEN version contains buggy programs generated by `DeepSeek-R1-Distill-Qwen-1.5B`, which is also used to generate our training data. ❷ UNSEEN version contains buggy programs generated by `Qwen3-14B`, which is never seen during training, and evaluates the generalizability of our models to different code generators. Please see Appendix A.2 for details of evaluation data.

**Metrics.** We extend the three standard metrics proposed in Prasad et al. (2025) for test harnesses. Specifically, ❶ **Good input (GI)** calculates the percentage of responses that have at least one bug-exposing input, *i.e.,* $\exists \boldsymbol{x}_i : f(\boldsymbol{x}_i) \neq g(\boldsymbol{x}_i)$. This metric purely measures the ability of the input generator. ❷ **Invalid test rate (ITR)** measures the percentage of responses where the ground-truth program fails, *e.g.,* tests that have invalid inputs or incorrect assertions. ❸ **True bug rate (TBR)** measures the percentage of responses that correctly expose the bug, *i.e.,* the ground-truth program passes the tests but the buggy program fails. This metric assesses the *overall performance*.

---

[1]Assertion errors in output verifier. All inputs still need to be valid, *i.e.,* do not trigger runtime errors on $g$.

Table 1: Performance on finding bugs (average of 8 runs). *: The model and training set are not released, so we compare with the number reported in the original paper. Note that the results of Qwen3-32B come from the original model without any fine-tuning.

| | MBPP+FIX (HARD) | | | LIVECODEBENCH | | | CODEFORCES | | |
|---|---|---|---|---|---|---|---|---|---|
| | GI ↑ | ITR ↓ | TBR ↑ | GI ↑ | ITR ↓ | TBR ↑ | GI ↑ | ITR ↓ | TBR ↑ |
| UTGen-32B* Prasad et al. (2025) | 56.1 | 40.8 | 34.7 | – | – | – | – | – | – |
| Qwen3-32B (Input/Output) | 56.4 | 10.1 | 49.3 | 56.7 | 5.1 | 54.8 | 79.9 | 21.6 | 67.1 |
| Qwen3-32B (Harness) | 78.7 | 11.9 | 68.6 | 69.1 | 15.5 | 55.1 | 80.4 | 33.9 | 54.8 |
| **Qwen3-4B** | | | | | | | | | |
| SFT (Input/Output) | 52.1 | 11.3 | 44.6 | 45.7 | 8.2 | 42.9 | 75.1 | 23.6 | 59.5 |
| SFT (Harness) | 78.1 | 17.7 | 62.9 | 60.4 | 23.7 | 42.1 | 82.5 | 46.9 | 46.0 |
| RL (Input/Output) | 82.5 | 13.9 | 72.7 | 68.4 | 9.9 | 65.1 | 89.8 | 21.0 | 72.2 |
| RL (Harness) | **84.4** | **13.0** | **74.1** | **79.1** | **9.5** | **69.9** | **91.8** | **19.1** | **74.4** |

For each input pair of problem and buggy program, we sample 8 responses and report the average performance of these 8 runs. We follow the official setting of Qwen3 to set the temperature at 0.6 and add a presence penalty of 1.5 Yang et al. (2025). The maximum generation length is set at 32,000.

**Number of Test Cases.** We allow each model response to contain one or more test cases. Concretely, for input-output testing, each model response could contain multiple pairs of input and expected output. For test harnesses, each response could contain multiple input generators, and each generator could further generate multiple test inputs. In our preliminary experiments, we observe that the number of test cases in each response significantly affects the performance (details in Appendix B.1). Thus, for the teacher model and SFT models, we report the performance of the best number of test cases. Namely, 1 test case per response for input-output testing and 5 test cases for test harnesses.[2] However, restricting the same number of test cases for all problems may be suboptimal. Therefore, during RL training, we allow the model to generate any number of test cases from 1 to 20, and the model learns the optimal number of test cases for each problem through training. The following section reports the performance of the above setting. In Appendix C.3, we further show results when controlling the number of test cases.

**Baselines.** We mainly compare with the baseline that generates input-output pairs for testing. For fair comparison, we conduct the same two-stage training as our method. Particularly, we use the same teacher model to generate an equal amount of SFT data, and we use the same reward in Eq. 1 for RL training. We additionally report the performance of directly prompting Qwen3-32B with both testing strategies. Finally, we compare with UTGen-32B Prasad et al. (2025), which also generates input-output pairs but is trained with only SFT without RL.

**Implementation Details.** We demonstrate the effectiveness of our framework on Qwen3-4B and Llama3.2-3B. For SFT, we train all models for 15 epochs and select the best checkpoint based on the validation performance. For RL, we leverage the Verl training framework Sheng et al. (2024) and train all models for 500 steps with a batch size of 128. Please see Appendix B.2 for detailed training hyperparameters. We parallelize the reward calculation for each rollout across all CPU cores, and on average, it takes 0.06 seconds to execute the test harnesses for each rollout during training. Appendix C.1 shows the detailed dynamics in RL training.

## 4.2 MAIN RESULTS

**Ability to Find Bugs.** Table 1 shows the performance of Qwen3-4B on finding bugs generated by models that have been seen during training. There are two observations from the table. **First**, our RL-trained model for test harness generation consistently outperforms the counterpart that generates input-output pairs. Specifically, it achieves better performance on all metrics across all benchmarks, demonstrating the benefits of test harness generation for both input generation and output verification. **Second**, both RL-trained small models surpass the 32B teacher models, which illustrates the effectiveness of our proposed two-stage training. Interestingly, although test harnesses initially underperform input-output generation on the teacher model and SFT models, our RL training

---

[2]If a response contains more test cases, we only evaluate the first 1 or 5 test cases.

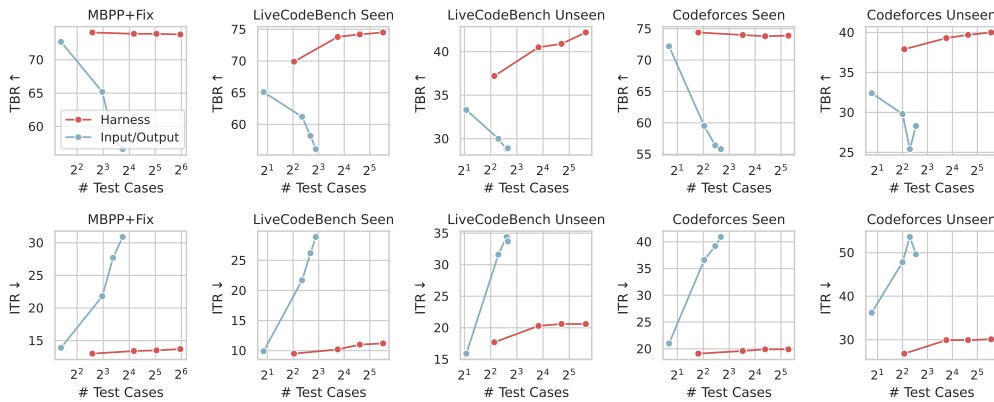

Figure 4: True bug rate (TBR) and invalid test rate (ITR) as the number of test cases increases.

unlocks their advantage and leads to better final performance. Appendix C.2 shows the results on `Llama3.2-3B`, which suggest that our method has better generalizability than input-output testing.

**Generalizability to Unseen Models.** We next evaluate our models' ability to debug for models that have never been seen during training. Specifically, we collect buggy programs generated by `Qwen3-14B`. These buggy programs are different from those in Table 1 in two ways: ① They are from an unseen model and thus may have different distributions for the bugs in the code. ② They are from a stronger model and pass more test cases, so they contain deeper logical bugs. Performance shown in Table 2 illustrates similar observations as Table 1. Particularly, our RL-trained test harness generators substantially outperform the model that generates input-output pairs. Moreover, our method

Table 2: Generalization to unseen models. The buggy code is sampled from `Qwen3-14B`, which is not seen during training.

| | LIVECODEBENCH | | | CODEFORCES | | |
|---|---|---|---|---|---|---|
| | GI ↑ | ITR ↓ | TBR ↑ | GI ↑ | ITR ↓ | TBR ↑ |
| **Qwen3-32B** | | | | | | |
| I/O | 25.0 | 8.3 | 23.0 | 43.2 | 20.1 | 31.5 |
| Harness | 36.8 | 20.4 | 22.4 | 61.6 | 36.3 | 32.3 |
| **Qwen3-4B** | | | | | | |
| SFT (I/O) | 19.4 | 12.4 | 17.3 | 35.7 | 25.4 | 23.1 |
| SFT (Har) | 34.1 | 34.5 | 16.3 | 59.1 | 45.2 | 25.3 |
| RL (I/O) | 37.0 | **15.9** | 33.3 | 53.0 | 36.2 | 32.4 |
| RL (Har) | **51.1** | 17.7 | **37.2** | **67.3** | **26.8** | **37.9** |

achieves larger improvements than Table 1. For instance, the relative improvement on CODEFORCES increases from 3.0% to 17.0%. The results show that our models can better generalize to unseen models. It also validates that the improvements of our method are not overfitting to a particular distribution of bugs.

**Scaling Number of Test Cases.** In the experiments above, we have limited each response to at most 20 test cases. We next investigate if we can further improve the performance by increasing the number of test cases in each response. Specifically, we employ different strategies to scale up the number of test cases for baselines and our method. For the baseline that generates input-output pairs, we directly change the instruction to the LLM to ask it to generate more test cases. For our method, since many input generators use random functions to generate inputs, we simply run the input generators multiple times with different random seeds to get more test inputs. Figure 4 shows the performance of the RL-trained models with respect to the number of test cases. As can be observed, when generating more test cases for the baseline method, the percentage of correctly identified bugs (TBR) drops significantly, and the amount of invalid tests (ITR) quickly increases, leading to a much worse performance. The observation confirms the limitations of hardcoded input-output pairs, since the probability of getting all test cases correct decays exponentially when the number of test cases increases. On the contrary, for our method, TBR consistently increases for three datasets and maintains the original value for the other two datasets, and ITR also demonstrates only a marginal increase. The results highlight two benefits of programmatic input generation and output verification: ❶ The input generator can *easily generate more inputs to increase the test coverage*; and ❷ The same output verifier can be reused for different inputs *without sacrificing the accuracy.*

**Using Feedback for Test-time Scaling.**
Given the superior bug-finding performance of
our model, we now explore its application to
improve code generation via test-time scaling.
Specifically, given a coding problem, we sample
8 candidate programs from an LLM and use
the test case generator to generate test cases
for each program. We collect all generated
test cases for the same problem and run them
against each candidate program. The program

Table 3: Best-of-8 performance on LIVE-
CODEBENCH where the code is selected based
on the execution results of the generated test cases.

|  | Code Generator | | |
|---|---|---|---|
|  | `Qwen3-4B` | `Qwen3-14B` | `Qwen3-32B` |
| Original pass@1 | 52.60 | 60.23 | 63.53 |
| `RL (I/O)` | 60.12 | 65.40 | 67.45 |
| `RL (Harness)` | **60.70** | **66.57** | **69.50** |

that passes the most test cases is selected as the final program. Table 3 shows the results on 341
problems of LIVECODEBENCH with three code generators. As can be observed, scaling with both
test case generators significantly improves the performance of the original LLM (original pass@1).
Furthermore, our model with test harnesses outperforms the input-output testing, demonstrating
its superior performance in judging code correctness. The results also confirm that our model's
improvements on finding bugs can be **translated into improved code generation**.

## 4.3 ADDITIONAL ANALYSES

**Diversity of Test Cases.** By programmatically
generating inputs, our model can potentially gen-
erate diverse inputs that would be difficult to
synthesize with hardcoding. We now verify this
by comparing the diversity of inputs generated
by the baseline and our models. Specifically, we
analyze the test cases generated for the programs
of `Qwen3-32B` in Table 3. We evaluate a sub-

Figure 5: Diversity of inputs for test cases gener-
ated by the two models.

|  | Unique ratio ↑ | Length range ↑ | Length std ↑ |
|---|---|---|---|
| `RL (I/O)` | 48.6 | 1.00 | 0.31 |
| `RL (Har)` | 77.1 | 8.90 | 2.69 |

set of 214 problems that take `stdin` as inputs. For fair comparison, we randomly downsample the
generated test cases so that the two models have the same number of test cases. We then calculate three
metrics for the two lists of inputs: ① Unique ratio: We calculate $\frac{\text{\# of unique inputs}}{\text{\# of total inputs}}$, where equality is de-
fined by string matching. ② Length range: We calculate $\log(\text{max\_length}+1) - \log(\text{min\_length}+1)$,
where $\text{max\_length}$ and $\text{min\_length}$ are the maximum and minimum lengths of the inputs. ③ Length
std: We calculate the standard deviation of the log of each input length. For all metrics, we compute
the value for each individual problem and take the average over all problems. Results in Table 5 show
that our method generates more diverse inputs and inputs with various lengths than the baseline.

**Performance across Difficulty Levels.** Section 4.2 reports aggregated performance across all
problems in a dataset. We next investigate if the improvement of our method is consistent across
problems with different difficulty levels. Figure 7 shows the detailed performance breakdown of the
baseline and our method. Specifically, on LIVECODEBENCH, we use the original difficulty categories.
On CODEFORCES, we split problems based on their ratings (HARD corresponds to problems with
ratings greater than 2400 and MEDIUM corresponds to problems with ratings greater than 1800). As
can be observed, while the performance of both methods degrades when problems become harder,
our method better maintains the performance compared to the baseline. The results indicate that
test harnesses can better generalize to difficult problems, verifying our motivation that input-output
testing is limited for complex problems.

**Distribution of Testing Strategies.** By programmatically generating inputs and validating outputs,
test harnesses allow models to have broader strategies for debugging. For example, we identify
two main ways models use to generate inputs, which are explicitly emphasized in our SFT data: ①
*Hardcoded*: models return a list of hardcoded inputs. ② *Dynamic*: models dynamically generates
inputs with code (*e.g.,* randomized inputs through random functions). Similarly, we identify three
ways models employ to validate a captured output: ① *Hardcoded*: models compare the output with a
hardcoded expected output. ② *Compare reference*: models implement a reference solution (*e.g.,* a
brute-force solution) and compare the output with that obtained from the reference solution. ③ *Check
invariant*: models check if the output satisfies specific invariants such as the length and range.

We prompt `Qwen3-32B` to classify the strategies used in each response (details in Appendix B.3).
Figure 8 shows the distributions of the input generator and the output verifier respectively. Specifically,

```
q: You are given an integer array nums. Select a subarray of nums such that: (1) All elements in the subarray are unique. (2) The
sum of the elements in the subarray is maximized. Return the maximum sum of such a subarray.

def generate_input_1():                        def check_output(input: Dict[str, Any], output: Any):
    """Hardcoded test case"""                      """Validates output for all cases"""
    return [{"nums": [-5,-3]}]                      nums = input["nums"]
                                                    if nums == [-5,-3]:
def generate_input_2():                                 assert output == -3
    """Dynamically generates test cases"""          else:
    input_list = []                                     all_negative = all(n < 0 for n in nums)
    for _ in range(3):                                  if all_negative:
        length = random.randint(1, 10)                      expected = max(nums)
        nums = [random.randint(-100, -1)                    assert output == expected
                for _ in range(length)]
        input_list.append({"nums": nums})
    return input_list
```

Figure 6: A sample output using a combination of strategies for input generators and output verifier.

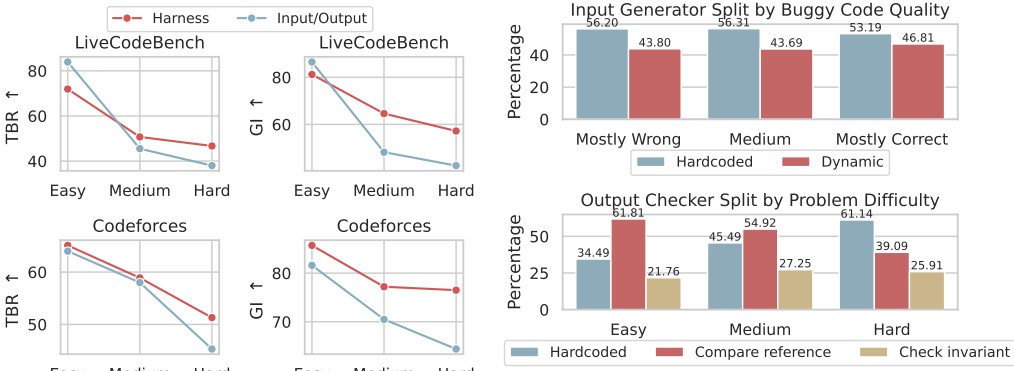

Figure 7: Performance across difficulty levels.

Figure 8: Distribution of testing strategies.

we report input generator strategies for buggy programs that are mostly wrong (pass less than 25% of test cases), medium (pass 25% to 75% of test cases), and mostly correct (pass greater than 75% of test cases). As can be observed, when the buggy program is mostly wrong and has obvious bugs, the model generates more hardcoded inputs. When the buggy program is more correct and contains bugs that are hard to identify, the model generates more dynamic inputs to increase test coverage.

Similarly, when the problem is easy, the model more often implements a reference solution for validation;[3] and when the problem becomes difficult, the model hardcodes more expected outputs. The observations demonstrate that the model can adapt its testing strategies to specific problems. Figure 6 shows an example where the model combines multiple strategies for output validation.

## 5 CONCLUSION AND FUTURE WORKS

We propose HarnessLLM, a pipeline for training LLMs for test harness generation. Through two-stage training of SFT followed by RLVR, we demonstrate that HarnessLLM outperforms its counterpart that generates input-output pairs. Additional experiments show that HarnessLLM exhibits better generalizability and benefits the code generation performance with test-time scaling.

One of the future directions is to explore methods that reduce the reliance on ground-truth programs. Currently, our method requires access to ground-truth programs during training to ensure the model generates valid test cases. In situations where the ground-truth programs are difficult to obtain, future works could explore directions such as using weaker oracles or generalizing models trained on simpler tasks with ground truths to more difficult tasks.

---

[3]An output verifier can use a combination of strategies, so the numbers do not add up to 100.

## 6 ETHICS STATEMENT

This work aims to enhance the reliability and robustness of AI-generated programs by developing improved methods for testing and debugging. However, while our method shows clear improvements over the baseline, it does not capture all bugs or provide any guarantees on the program's correctness. Our experiments show that some bugs remain hidden and some correct programs may be mistakenly flagged. Therefore, users should remain cautious when interpreting the execution results of our generated test cases. We advise that any use of this system in high-stakes environments be accompanied by additional verification and human oversight.

## 7 REPRODUCIBILITY STATEMENT

We have taken the necessary steps to ensure the reproducibility of our results. Specifically, Section 4.1 discusses the general experiment settings in our paper. Appendix A provides the detailed steps to collect the training and evaluation datasets. Finally, Appendix B lists the implementation details of our method and baselines, including training hyperparameters and evaluation details.

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

# A  DATASET CONSTRUCTION

## A.1  TRAINING DATA

To train LLMs for test case generation, we collect data in the triplets of problem description $q$, buggy program $f$, and ground-truth program $g$. We consider Python programs in this paper. We source such triplets from existing coding datasets, including TACO Li et al. (2023), SYNTHETIC-1 Intellect (2025), LeetCode Xia et al. (2025), and Codeforces MatrixStudio (2025). These datasets come with the problem description, a ground-truth program, and a list of ground-truth test cases. We use the following three steps to collect data:

❶ Filter ground-truth programs: We run the given ground-truth program $g$ on all test cases and only keep problems where $g$ passes all test cases.

❷ Generate buggy programs: We sample candidate programs from `Qwen2.5-Coder` 1.5B, 3B, 7B Hui et al. (2024), and `DeepSeek-R1-Distill-Qwen-1.5B` DeepSeek-AI (2025). We sample 8 programs from each model and run the programs on all ground-truth test cases. We only keep programs that pass at least one test case but not all test cases,

Table 4: Statistics of our training data.

|  | Statistic |
| --- | --- |
| # triplets for RL | 12,043 |
| # unique problems for RL | 7,748 |
| # triplets for SFT | 6,805 |
| # unique problems for SFT | 4,383 |
| # responses for SFT | 15,619 |

resulting in partially correct programs. If there are multiple candidates that satisfy the requirement, we use the two that pass the most test cases, which makes it harder to find bugs.

❸ Decontamination: We decontaminate training data against all evaluation benchmarks based on the problem description.

We use all collected data for RL training and a subset of data for SFT, ensuring that models see new data during RL training. Table 4 shows the statistics of our training set. Specifically, the dataset contains two types of problems: standard input/output problems that read from `stdin` and return to `stdout`, as well as functional problems that implement a function in Python. Since the number of functional problems is small, we create two versions for each functional problem, where one contains a few example input-output pairs in the description, and the other does not.

**SFT Data.**  To collect SFT data, we use the rejection sampling technique Touvron et al. (2023). Specifically, we prompt `Qwen3-32B` to generate 6 responses for each pair of description and buggy program. Figures 10 and 11 show the prompt we use for input-output testing and test harnesses, respectively. Particularly, for harness generation, we encourage the model to use diverse strategies to validate outputs, such as checking specific invariants and comparing with a brute-force solution, which is similar to the strategy used in prior works Zhang et al. (2023b). We run generated test cases on both ground-truth program $g$ and buggy program $f$ and only keep responses where $g$ passes the test but $f$ does not. We keep the amount of SFT data the same for input-output testing and harness testing.

## A.2  EVALUATION DATA

We evaluate on three popular code generation datasets: MBPP+ Austin et al. (2021); Liu et al. (2023), LIVECODEBENCH Jain et al. (2025), and CODEFORCES Penedo et al. (2025). Although these datasets are designed for code generation tasks, we convert them into bug-finding tasks following the procedure in Section A.1.

Specifically, for LIVECODEBENCH, we use problems from 2024/10 to 2025/4. For CODEFORCES, we use samples in the test split. For both datasets, we use correct public submissions as the ground-truth

Table 5: Statistics of evaluation datasets.

|  | # data |
| --- | --- |
| MBPP+FIX (HARD) | 141 |
| LIVECODEBENCH SEEN | 76 |
| LIVECODEBENCH UNSEEN | 93 |
| CODEFORCES SEEN | 100 |
| CODEFORCES UNSEEN | 84 |

program, after rerunning and filtering the submissions on all test cases.

Table 6: True bug rate (higher is better) of input-output-based testing with `Qwen3-32B` when only evaluating the first $k$ generated test cases. We use the SEEN version of LIVECODEBENCH and CODEFORCES.

|  | MBPP+ | LIVECODEBENCH | CODEFORCES |
|---|---|---|---|
| $k = 1$ | 49.3 | 54.8 | 67.1 |
| $k = 3$ | 59.0 | 54.6 | 53.2 |
| $k = 5$ | 57.4 | 53.6 | 42.5 |
| $k = 10$ | 54.6 | 51.3 | 39.5 |

Table 7: True bug rate (higher is better) of test harnesses with `Qwen3-32B` when only evaluating the first $k$ generated test cases. We use the SEEN version of LIVECODEBENCH and CODEFORCES.

|  | MBPP+ | LIVECODEBENCH | CODEFORCES |
|---|---|---|---|
| $k = 3$ | 66.6 | 48.5 | 57.9 |
| $k = 5$ | 68.6 | 55.1 | 54.8 |
| $k = 10$ | 67.7 | 53.8 | 48.6 |
| $k = 20$ | 67.3 | 53.3 | 44.2 |

For MBPP+, we directly use the split MBPP+FIX (HARD) in `UTGen-32B` Prasad et al. (2025), which is collected similarly to the above procedure. Particularly, we notice the problem descriptions in MBPP+ are overly simplified and without clear input specifications (*e.g., 'Write a function to find the length of the longest palindromic subsequence in the given string'*, without specifying that the input string should be non-empty). We thus use `Qwen3-32B` to add an input specification to the problem (detailed prompt in Figure 12). To make sure the ground-truth program $g$ matches the description after modification, we further prompt `Qwen3-32B` to adapt the original $g$ to the new description (detailed prompt in Figure 13). Finally, we filter the modified ground-truth programs and only keep those that pass the original ground-truth test cases.

Table 5 lists the statistics of all evaluation benchmarks.

## B    IMPLEMENTATION DETAILS

### B.1    NUMBER OF TEST CASES

For the teacher model and SFT models, we observe that the number of test cases in a response significantly affects the final performance. For example, although we allow models to generate multiple test cases in each response, Tables 6 and 7 show that the performance of `Qwen3-32B` can vary significantly if we only evaluate the first $k$ test cases. Both methods' performance improves as we evaluate on fewer test cases, especially for input-output-based testing. This confirms the observations in Figure 4, where the performance of input-output testing quickly drops when generating more test cases. Based on these results, for the teacher model and SFT models of input-output testing, we report the performance when $k = 1$. For test harnesses, we report the performance when $k = 5$.

For the RL models, we observe that the models automatically find a good number of test cases to generate. For instance, the RL trained `Qwen3-4B` model for input-output testing generates 1.96 test cases in each response on average. Thus, we allow the model itself to determine the number of test cases, and we only restrict the maximum test cases at 20.

### B.2    TRAINING HYPERPARAMETERS

We run all experiments on 16 NVIDIA H100 GPUs. The RL training for our model takes around 1,500 GPU hours. The RL training for the input-output baseline takes around 1,150 GPU hours. Table 8 lists the hyperparameters for SFT and RL training. Note that we use the same hyperparameters for all models.

### B.3    CLASSIFYING TESTING STRATEGIES

We prompt `Qwen3-32B` to identify specific testing strategies used by our model. Specifically, given the generated harness code, we ask the model to identify strategies used in each input generator and output verifier. The detailed prompts are listed in Figures 14 and 15.

Table 8: Training hyperparameters. The same hyperparameters are used for all models.

| SFT Training | |
|---|---|
| # Epochs | 15 |
| Batch size | 96 |
| Learning rate | $1e{-}5$ |
| LR scheduler | cosine |

| RL Training | |
|---|---|
| # Steps | 500 |
| Batch size | 128 |
| # Rollouts per question | 8 |
| Learning rate | $1e{-}6$ |
| LR scheduler | None |
| Max response length | 16,384 |

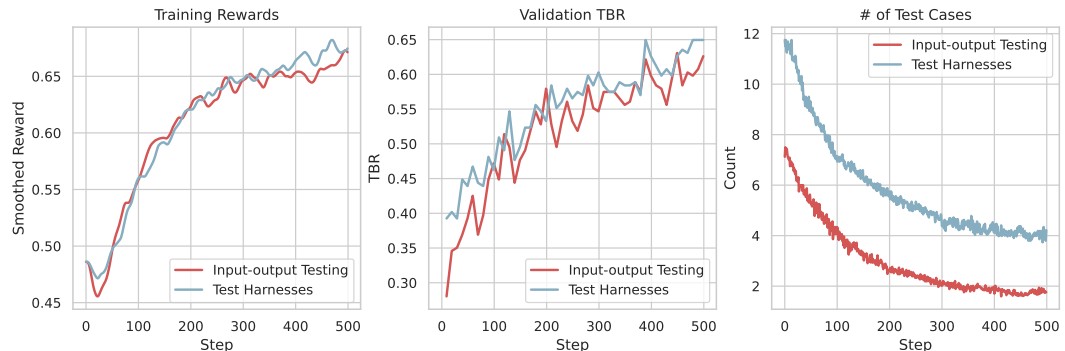

Figure 9: Dynamics of smoothed training rewards, true bug rage (TBR ↑) on validation set, and the number of generated test cases throughout RL training.

| | MBPP+Fix (Hard) | | | LCB Seen | | | CF Seen | | | LCB Unseen | | | CF Unseen | | |
| --- | --- | --- | --- | --- | --- | --- | --- | --- | --- | --- | --- | --- | --- | --- | --- |
| | GI ↑ | ITR ↓ | TBR ↑ | GI ↑ | ITR ↓ | TBR ↑ | GI ↑ | ITR ↓ | TBR ↑ | GI ↑ | ITR ↓ | TBR ↑ | GI ↑ | ITR ↓ | TBR ↑ |
| RL (I/O) | 71.3 | 37.3 | **45.3** | 47.0 | 42.3 | 30.3 | 67.6 | 53.9 | **32.8** | 21.0 | 61.8 | 8.3 | 37.4 | 66.2 | 10.1 |
| RL (Har) | **77.9** | 37.3 | 42.6 | **76.5** | **33.9** | **31.4** | **81.4** | **43.8** | 29.9 | **59.9** | **39.9** | **17.7** | **73.4** | **43.3** | **21.6** |

Table 9: Performance of `Llama3.2-3B` on finding bugs (average of 8 runs). `I/O`: input-output testing. `Har`: test harnesses.

## C  ADDITIONAL RESULTS

### C.1  TRAINING DYNAMICS

Figure 9 shows the dynamics of RL training for both methods. As can be observed, our method consistently achieves a higher TBR on the validation set than the baseline. Moreover, both methods generate fewer test cases as the training progresses, approaching the optimal number in Section B.1. This indicates that the models are learning the best number of test cases for generation.

### C.2  RESULTS ON LLAMA

Table 9 shows the performance when training on `Llama3.2-3B` model. As can be observed, our model for test harnesses achieves comparable performance with input-output testing on the SEEN version of the datasets. However, it significantly outperforms the input-output testing when evaluated on the UNSEEN version, *e.g.*, a relative improvement over 110% in TBR on LIVECODEBENCH. The results indicate that input-output testing has the risk of overfitting to a particular distribution of bugs, whereas test harnesses have better generalizability.

### C.3  PERFORMANCE UNDER CONTROLLED NUMBER OF TEST CASES

Our experiments in Section 4.2 demonstrate that different methods should generate different numbers of test cases for the best performance. Particularly, for input-output testing, models usually have better performance when generating fewer test cases, since more test cases lead to a higher probability that one of the test cases is wrong. By contrast, for test harnesses, performance can be further improved by scaling up the number of test cases, which increases the test coverage without sacrificing accuracy. Nevertheless, in the following section, we also report the performance when controlling the number of test cases.

Specifically, we repeat the experiments in Tables 1 and 2 but instruct the input-output testing model to only generate a single test case in each response. For test harnesses, we also prompt the model to generate a single input generator. Then, to get more test cases, we sample multiple responses for input-output testing and run the same input generator multiple times with different random seeds for test harnesses. Table 10 presents the performance under this controlled setting. As can be observed, our method consistently outperforms input-output testing, and the gap becomes larger

| | MBPP+Fix (Hard) | | | LCB Seen | | | CF Seen | | | LCB Unseen | | | CF Unseen | | |
|---|---|---|---|---|---|---|---|---|---|---|---|---|---|---|---|
| | GI ↑ | ITR ↓ | TBR ↑ | GI ↑ | ITR ↓ | TBR ↑ | GI ↑ | ITR ↓ | TBR ↑ | GI ↑ | ITR ↓ | TBR ↑ | GI ↑ | ITR ↓ | TBR ↑ |
| **Repeat Once** | | | | | | | | | | | | | | | |
| RL (I/O) | 80.5 | 11.6 | 73.8 | 67.4 | 9.7 | 65.1 | 87.8 | 16.2 | 75.6 | 36.0 | 18.7 | 33.9 | 49.1 | 30.7 | 34.2 |
| RL (Har) | 83.6 | 12.1 | **74.9** | 75.0 | 7.9 | **68.4** | 90.2 | 16.1 | **76.9** | 43.3 | 13.6 | **37.1** | 58.3 | 21.4 | **34.5** |
| **Repeat 5 Times** | | | | | | | | | | | | | | | |
| RL (I/O) | 86.7 | 22.3 | 69.5 | 79.9 | 23.7 | 63.2 | 97.2 | 34.8 | 63.8 | 54.8 | 46.5 | 31.7 | 75.0 | 56.2 | 31.8 |
| RL (Har) | 83.8 | 12.3 | **75.0** | 75.0 | 7.9 | **68.4** | 90.8 | 16.1 | **77.4** | 49.5 | 14.5 | **40.9** | 61.9 | 22.6 | **34.5** |
| **Repeat 20 Times** | | | | | | | | | | | | | | | |
| RL (I/O) | 88.7 | 34.0 | 60.3 | 86.8 | 42.1 | 51.3 | 100.0 | 56.0 | 44.0 | 64.5 | 72.0 | 18.3 | 86.9 | 79.8 | 16.7 |
| RL (Har) | 83.5 | 12.5 | **74.6** | 77.6 | 7.9 | **71.1** | 90.8 | 16.4 | **77.1** | 52.2 | 14.9 | **42.5** | 63.1 | 23.8 | **35.7** |

Table 10: Bug-finding performance of `Qwen3-4B` when controlling the number of test cases. `I/O`: input-output testing. `Har`: test harnesses.

when increasing the number of test cases. These results demonstrate the consistent improvements of the proposed test harness.

Moreover, we also rerun the test-time scaling experiment in Table 3. Here, we further restrict each input generator to have a single test input, thus ensuring the two methods have the same number of test cases for each candidate program. For both methods, we sample 5 responses for each candidate program to obtain more test cases. The results in Table 11 show that our method surpasses the baseline in most settings. Particularly, it significantly outperforms the baseline when the strongest `Qwen3-32B` is used as the code generator, demonstrating the superior generalizability and potential weak-to-strong generalization of our method.

Table 11: Best-of-8 performance on LIVE-CODEBENCH where the code is selected based on the execution results of the generated test cases.

| | Code Generator | | |
|---|---|---|---|
| | Qwen3-4B | Qwen3-14B | Qwen3-32B |
| Original pass@1 | 52.60 | 60.23 | 63.53 |
| **1 test case per program** | | | |
| RL (I/O) | 60.41 | **65.10** | 65.98 |
| RL (Harness) | **60.70** | 64.81 | **68.33** |
| **5 test cases per program** | | | |
| RL (I/O) | **61.88** | 67.16 | 68.04 |
| RL (Harness) | 61.00 | **67.74** | **72.14** |

```
Given a problem statement and a Python program that aims to solve it, your
task is to **write test cases** that uncover any potential bugs.

### **Task Overview**

You should output a JSON object that contains a list of test cases for the
provided program. Each test case should include:
1. **input_str**: The exact text to feed into stdin.
2. **expected_output**: The exact text the program should print.

We will run each test by feeding `input_str` into the program and comparing
its stdout against `expected_output`.

### **Required Format**

```json
[
  {
    "input_str": "input 1",
    "expected_output": "output 1"
  },
  {
    "input_str": "input 2",
    "expected_output": "output 2"
  }
  // ... up to 20 test cases total
]
```

### **Constraints**

* Generate **1-20** test cases.
* Don't include comments or extra fields in the JSON.
* Each input_str and expected_output must be a valid JSON string.

The problem is as follows:
{description}

And the program is as follows:
```python
{target_code}
```
```

Figure 10: Prompt used for input-output-based testing. Note that this prompt assumes the program reads input from stdin.

Given a problem statement and a Python program that aims to solve it, your
task is to **write a test harness** that uncovers any potential bugs.

### **Task Overview**

You will deliver **a single** code block to define functions that can be run
 by our framework to generate inputs, run the program, and validate its
outputs.
Consider two categories of test cases:
- **Hardcoded cases**: Manually crafted input-output pairs that expose known
 or likely bugs.
- **Dynamic cases**: Programmatically generated inputs that stress-test the
implementation (e.g., randomized, combinatorial, large or edge-case inputs).

### **Required Functions**

```python
from typing import List

def generate_input_1() -> List[str]:
    """
    Return between 1 and 4 valid input strings, each a complete stdin
    payload for the target program.
    Consider the following strategies:
       - Manually craft inputs that expose bugs.
       - Dynamically generate randomized, combinatorial, large, or edge-case
       inputs for stress testing.
    """
    # Your code here
    return input_list

def generate_input_2() -> List[str]:
    """
    Another function to return between 1 and 4 valid input strings.
    Employ a different strategy than previous input generation functions.
    """
    # Your code here
    return input_list

# You may add up to 3 more functions named generate_input_3(),
generate_input_4(), etc.

def check_output(generated_input: str, captured_output: str) -> None:
    """
    Validate the output for a single generated input.
    Inputs:
        - generated_input: The input string passed to the target program.
        - captured_output: The exact stdout produced by the target program.

    Hints: When exact outputs are hard to predict, avoid asserting them.
    Instead, consider:
       - Check key properties or invariants, e.g., output is sorted, has
       correct length, matches a pattern, has correct value ranges, etc.
       - Compare against a simple brute-force implementation
    """
    # Your code here
```

### **Execution Flow**

1. The framework calls generate input functions to obtain a list of test
strings.
2. For each string:
   * It runs the target program with that string on stdin.
   * Captures stdout into `captured_output`.
   * Calls `check_output(generated_input, captured_output)`.
3. If any assertion fails, the test suite reports an error.

### **Constraints**

```
* Provide one contiguous block of Python code that defines all required/
optional functions. Do not invoke the functions yourself-only define them.
* Define up to 5 input generation functions, each returning between 1 and 4
inputs.
* The dynamic input functions must employ diverse strategies to generate
inputs. Avoid generating inputs with the same logic or from the same
distribution.
* Runtime limit per check_output call: 5 seconds.

The problem is as follows:
{description}

And the program is as follows:
```python
{target_code}
```
```

Figure 11: Prompt used for test harnesses generation. Note that this prompt assumes the program reads input from stdin.

```
Given the following coding problem and a corresponding solution, improve the
 problem description by adding input specifications. Include details such as
:
- Valid input types (e.g. "integer", "string", "list of floats").
- Reasonable value ranges (e.g. "0 <= n <= 1000").
- Format constraints (e.g. "no empty strings", "no null/None values").

Do not change the original requirements or add example cases, just append
the specifications.

Problem:
{problem}

Code:
```python
{code}
```
```

Figure 12: Prompt used for adding input specifications on MBPP+.

```
Given the following coding problem and a corresponding solution, decide
whether the solution contains a bug or not. If yes, rewrite the code to fix
the bug. Remember to look for edge cases where the code fails to handle.

Problem:
{problem}

Code:
```python
{code}
```

Output your answer in the following format:
```python
fixed_code
```
where fixed_code is the rewritten code that fixes the bug. If the code is
correct, just return the original code without any changes.
```

Figure 13: Prompt used for adapting the ground-truth programs to the new descriptions on MBPP+.

```
Given the following code snippet for a test harness, determine the strategy
used in each `generate_input` function.

Code:
```python
{code}
```

Select from the following options:
- hardcoded: the function returns hardcoded inputs.
- dynamic: the function generates inputs dynamically, e.g., random sampling,
 or combinatorial generation.

Think about the code step by step and then output your final answer in the
following format:
```json
<used strategies>
```
where <used strategies> is a list of the strategies used in each function.

Notes:
- The list should have the same length as the number of `generate_input`
functions in the code.
- If a function uses a combination of the above strategies, select the
dominant strategy.
```

Figure 14: Prompt used for identifying strategies in input generators.

```
Given the following code snippet for a test harness, determine the
strategies used in the `check_output` function.

Code:
```python
{code}
```

Select from the following options:
- reference implementation: the function compares the output with a
reference implementation, e.g., a brute-force solution, or a correct
implementation.
- invariant checking: the function checks whether the output satisfies
certain invariants or properties, e.g., whether the output is sorted, or
whether the output has valid types and lengths.
- hardcoded: the function compares the output with hardcoded expected
outputs.

Think about the code step by step and then output your final answer in the
following format:
```json
<used strategies>
```
where <used strategies> is a list of the strategies used in the function.

Notes:
- If the function uses a combination of the above strategies, return a list
containing all the strategies used, e.g., ["reference implementation", "
invariant checking"].
- If the function does not contain any of the above strategies, return an
empty list [].
```

Figure 15: Prompt used for identifying strategies in the output verifier.

