# OpenReview forum: "HarnessLLM: Automatic Testing Harness Generation via Reinforcement Learning"
_ICLR.cc/2026/Conference — ICLR 2026 Conference Withdrawn Submission_

### Official Review · Reviewer_1wej · 2025-10-27

**Soundness:** 2
**Presentation:** 2
**Contribution:** 1
**Rating:** 2
**Confidence:** 4

**Summary:**

This paper propose HarnessLLM, a two-stage training pipeline that enables LLMs to write harness code for testing. Particularly, LLMs generate code that synthesizes inputs and validates the observed outputs, allowing complex test cases and flexible output validation such as invariant checking. To achieve this, authors train LLMs with SFT followed by RLVR with a customized reward design.

**Strengths:**

1.The writting is good and easy to follow
2.The paper presents a clear motivation

**Weaknesses:**

1.The proposed method appears heavily dependent on the base model's inherent capabilities, and given that relatively small models (e.g., Qwen3-4B) were selected for experimentation, it remains unclear whether the approach would yield comparable performance improvements when applied to larger, more capable models.
2.The experimental evaluation is insufficiently comprehensive: the comparison with larger models is limited, the evaluation datasets contain relatively small sample sizes that raise questions about statistical robustness, the performance gains over the input-output baseline are marginal in some cases, and additional experiments differentiating this work from existing approaches would strengthen the empirical validation.
3.The related work section lacks thoroughness, as the paper does not adequately position itself within the broader landscape of automated test generation, LLM-based software engineering tools, and property-based testing methodologies.
4.The novelty of the contribution is limited, as the core idea of generating executable test code rather than test cases has been explored in prior work on property-based testing and programmatic test generation.

**Questions:**

1.What methodology do the authors employ to filter and select suitable training data from the generated test cases?
2.To what extent can the generated test data surpass the intrinsic performance boundaries of the underlying model?

---

> ### Author Response · Authors · 2025-11-27
>
> We thank the reviewer for the valuable comments and address each concern below.
>
> **W1: The proposed method depends on the base model’s capabilities. It’s unclear if the improvements can generalize to larger models**
>
> To examine generalization across model capacities and architectures, we conduct experiments with **two different base models**: Qwen3-4B and LLaMA3.2-3B. Although Qwen3-4B is stronger than LLaMA3.2-3B, we observe consistent improvements over the input-output baseline on both models. This suggests that our two-stage training pipeline is not tailored to a single backbone model, but instead improves test generation for different models. Beyond small and medium models, we are currently training a larger LLM (7B) and will update the results if the training completes in time.
>
> **W2: The evaluation is insufficient: comparison with larger models is limited, and evaluation datasets are relatively small**
>
> We follow the reviewer’s suggestions to compare with larger models. In particular, we evaluate openai/gpt-oss-120b (frozen) on the two unseen versions of the datasets. The results in the following table show that this model significantly outperforms Qwen3-32B. Additionally, harness generation with gpt-oss also outperforms the input-output baseline, indicating the effectiveness of harness generation when the model is strong. Moreover, our RL-trained 4B model is the second-best model across the table, only behind the gpt-oss with harness generation, which shows the benefits of our two-stage training.
> |  | LCB Unseen |  |  | CF Unseen |  |  |
> |---|---|---|---|---|---|---|
> |  | GI ↑ | ITR ↓ | TBR ↑ | GI ↑ | ITR ↓ | TBR ↑ |
> | I/O (Qwen3-32B) | 25.0 | 8.3 | 23.0 | 43.2 | 20.1 | 31.5 |
> | Harness (Qwen3-32B) | 36.8 | 20.4 | 22.4 | 61.6 | 36.3 | 32.3 |
> | I/O (gpt-oss-120B) | 32.4 | 5.2 | 30.6 | 53.3 | 19.0 | 41.7 |
> | Harness (gpt-oss-120B) | 53.5 | 17.5 | 42.3 | 77.7 | 32.6 | 48.1 |
> | I/O (RL-4B) | 37.0 | 15.9 | 33.3 | 53.0 | 36.2 | 32.4 |
> | Harness (RL-4B) | 51.1 | 17.7 | 37.2 | 67.3 | 26.8 | 37.9 |
>
> Regarding the size of evaluation datasets, although each evaluation dataset is moderately sized, we sample 8 responses per question and report the average performance, so the **variance of this average performance is actually small**. For example, on the seen version of Codeforces, where the gap of true bug rate (TBR) is small (our method has 74.4 and input-output testing has 72.2), the standard deviation of the average is 0.85 for our method and 0.42 for input-output testing, which suggests a non-trivial difference.
>
> Finally, we have evaluated on **two additional versions of the datasets**, where the buggy programs are sampled from openai/gpt-oss-20b, which is from a completely different model family. The results in the following table demonstrate that our method outperforms the baseline across these harder test sets. This suggests that the observed improvements are stable across model families and dataset difficulty levels.
> |  |  | LCB |  |  | CF |  |
> |---|---|---|---|---|---|---|
> |  | GI ↑ | ITR ↓ | TBR ↑ | GI ↑ | ITR ↓ | TBR ↑ |
> | I/O (RL) | 24.1 | 20.9 | 20.7 | 43.5 | 29.3 | 28.9 |
> | Harness (RL) | 39.3 | 21.5 | **24.7** | 63.7 | 26.6 | **30.5** |
>
> **W3: Missing related works on automated test generation, LLM-based software engineering tools, and property-based testing**
>
> Thanks for the suggestion. We will revise our related works section to incorporate prior works in the mentioned areas and discuss the relations between our work and existing works.
>
> **W4: The novelty is limited**
>
> We want to clarify that the contribution of our work is not only about introducing the concept of harness generation for debugging. In addition, we propose a two-stage training pipeline, which is empirically verified to improve the model’s test generation capability. Furthermore, we collect SFT and RL datasets, which can be used by the community to train other code LLMs. For example, most of the prior works on LLM-powered property-based testing [1-3] prompt a frozen LLM to generate property tests. In our experiments, we show that zero-shot prompting on a small model fails to generate effective test harnesses, but our training pipeline significantly improves the base model performance.
>
> [1] Vikram et al., Can Large Language Models Write Good Property-Based Tests?
>
> [2] He et al., Use Property-Based Testing to Bridge LLM Code Generation and Validation.
>
> [3] Bose et al., ​​From Prompts to Properties: Rethinking LLM Code Generation with Property-Based Testing.

---

> > ### Author Response · Authors · 2025-11-27
> >
> > **Q1: How do the authors filter and select suitable training data from generated test cases?**
> >
> > To filter generated test cases for SFT training, we adopt the standard rejection sampling technique. Particularly, we use the teacher model (Qwen3-32B) to sample 6 responses for each pair of input problem and buggy implementation. Each response may contain multiple test cases, and we execute all test cases against the ground-truth and buggy programs, respectively. We only keep the responses where the ground-truth program passes all test cases, but the buggy program fails on at least one of the test cases. This makes sure that the remaining responses contain effective test harnesses that correctly identify the bugs in the input program.
> >
> > **Q2: To what extent can the generated test data surpass the performance boundary of the base model?**
> >
> > We have constructed and evaluated on datasets of increasing difficulty:
> > - The first version uses buggy programs generated by DeepSeek-R1-Distill-Qwen-1.5B, which is also used for generating our training data.
> > - The second version contains buggy programs generated by Qwen3-14B, which is a much stronger model and is never seen during training.
> > - A third, even harder version uses buggy programs from openai/gpt-oss-20b (results shown in our response to **W2**).
> >
> > Empirically, the true bug rate (TBR) drops as we move from version 1 to 3, confirming that using stronger generators yields more challenging test sets. However, across all these difficulty levels, our trained harness model consistently outperforms the input-output baseline, demonstrating its effectiveness.

---

### Official Review · Reviewer_ThXT · 2025-10-28

**Soundness:** 2
**Presentation:** 3
**Contribution:** 2
**Rating:** 4
**Confidence:** 4

**Summary:**

The paper proposes HarnessLLM, a two-stage training approach for effective test generation. The main insight of HarnessLLM is to teach LLMs to write harness code for testing. The generated harness code first synthesizes diverse inputs and then generate validation function to check the correctness of the inputs and outputs. The experiments show that the proposed approach can improve the bug finding capabilities of test generation.

**Strengths:**

The paper targets an important problem (test generation)

**Weaknesses:**

The paper shapes the test generation problem in a simple and less practical scenario. In particular, the test generation scenario should assume there is a problem statement (which serves as the specification for the correct code). However, such an assumption is not always held, actually there are no problem statements or natural language-described specification for real-world software systems at most cases. Therefore, the application scenario of the proposed approach is narrow and not realistic. And it is almost infeasible to apply the approach to a real-world software repository.

The motivation of the approach design needs more justification. The paper proposes to train models to generate better test inputs and a better verifier (to check the outputs). The verifier is supposed to distinguish the behavior between the correct ground-truth program and the buggy program. If that is the case, what is the difference between generating a verifier and directly generating the correct program? That is, generating the correct program for the given problem statement can always serve as the verifier. Therefore, the idea of generating the verifier is essentially generating the code program (i.e., a code generation problem), but in a less direct way.


The paper misses state-of-the-art test generation techniques, especially those based on LLMs. In fact, generating test cases have been widely explored in existing literature, including both generating diverse test inputs and generating test oracles. Many of them are not included and discussed in the paper[a,b,c].

[a] Advancing Code Coverage: Incorporating Program Analysis with Large Language Models. TOSEM

[b] CoverUp: Effective High Coverage Test Generation for Python. FSE’25

[c] CODAMOSA: Escaping Coverage Plateaus in Test Generation with Pre-trained Large Language Models. ICSE’23


The proposed approach uses prompts to generate diverse dynamic test inputs. But it is unclear what is the benefits of the proposed prompt, especially over existing test input generation techniques (e.g., search-based test input generation like Evosuite or random fuzzing techniques like AFL). Currently, there is no justification on the effectiveness and strength of such an approach design.

The evaluation can be further enhanced with including more diverse programming languages (e.g., Java) and more studied LLMs.

In addition, the current subjects are too naïve for test generation tasks given the limited complexity of function-level code snippets. It is common practice to study on real-world projects.

**Questions:**

1.	What is the difference between generating a verifier and directly generating the correct code?

2.	How does the approach compare to more representative LLM-based test generation techniques?

3.	How could the approach be applied to real-world software repositories where there are no natural-language specifications?

4.	Why use prompts to generate diverse test inputs? What is the benefits over existing search-based or fuzzing techniques?

---

> ### Author Response · Authors · 2025-11-27
>
> We thank the reviewer for the valuable comments and address each concern below.
>
> **W1: The paper studies a simplified scenario. It is difficult to obtain problem statements for real-world software systems**
>
> We agree that our evaluation setting, i.e., function-level test generation given a problem statement, is simplified compared to end-to-end testing of large repositories. However, our intention is to study **a clean abstraction of the LLM’s core ability to generate effective test cases**, rather than to propose a full-stack testing pipeline.
>
> Specifically, recent works on repo-level debugging often adopt an agentic, multi-stage framework, where specialized agents are created for particular sub-tasks in the whole pipeline [1-4]. For example, some agents are used to first locate candidate code units responsible for the bugs, and other agents then generate test cases targeting each localized unit. Although the whole repository might not contain problem statements for each function, once a function is identified to contain potential bugs, we can use abstract syntax tree parsers to extract the dependencies and build a relevant context. Within this context, the problem statement and specification of the function can be analyzed, and test generation reduces to a similar setting studied in this paper. Our evaluation therefore serves as an abstraction of the test generation component inside a larger system, isolating the model’s test case generation ability without the potential confounding factors introduced by the whole pipeline. We will make this scope explicit in the revised paper.
>
> **W2: What’s the difference between generating a verifier and generating a correct program for the given problem**
>
> We agree that generating a correct reference implementation and using it as an oracle is a legitimate approach. In fact, our model does learn to generate a correct program and use it to produce expected outputs (please see examples below). However, the difference is that our model does more than this and covers **more diverse strategies for verifying outputs**. For example, our model can verify outputs by checking the required invariants and properties, which allows it to verify outputs even if it does not know the correct implementation. Figure 8 in our paper confirms that our model can **adapt its verifying strategy to the given problem**. When the problem is easy, the model more often generates correct programs to produce expected outputs, but when the problem is difficult, the model more often uses invariant-based checking or directly hardcodes the expected outputs. This flexibility gives it advantages over the method that always tries to generate a correct program.
>
> [1] Tang et al., Co-PatcheR: Collaborative Software Patching with Component(s)-specific Small Reasoning Models.
>
> [2] Chen et al., ChatUniTest: A Framework for LLM-Based Test Generation.
>
> [3] Pan et al., ASTER: Natural and Multi-language Unit Test Generation with LLMs.
>
> [4] Lops et al., A System for Automated Unit Test Generation Using Large Language Models and Assessment of Generated Test Suites.

---

> > ### Author Response · Authors · 2025-11-27
> >
> > Below are examples of our model using different strategies to verify outputs.
> >
> > Here is an example where the model checks the properties of the output:
> > ```
> > def check_output(generated_input: str, captured_output: str) -> None:
> >     """
> >     Validate that every number from 1 to n appears exactly twice in the output.
> >     """
> >     n = int(generated_input.strip())
> >     expected_length = 2 * n
> >     output_list = list(map(int, captured_output.strip().split()))
> >
> >     # Ensure the output has the correct length
> >     assert len(output_list) == expected_length, f"Output length mismatch: expected {expected_length}, got {len(output_list)}"
> >
> >     # Count occurrences of each number
> >     freq = [0] * (n + 1)
> >     for num in output_list:
> >         assert 1 <= num <= n, f"Invalid number detected: {num} (out of bounds for n={n})"
> >         freq[num] += 1
> >
> >     # Verify each number appears exactly twice
> >     for i in range(1, n + 1):
> >         assert freq[i] == 2, f"Number {i} appears {freq[i]} times instead of 2"
> > ```
> >
> > Here is an example where the model implements a brute-force function to produce expected outputs:
> > ```
> > def check_output(generated_input: str, captured_output: str) -> None:
> >     """
> >     Validates the output for small cases via brute-force comparison.
> >     """
> >     parts = generated_input.strip().split()
> >     n = int(parts[0])
> >     p = int(parts[1])
> >     k = int(parts[2])
> >
> >     # Brute-force for small n and p
> >     if n <= 4 and p <= 7:
> >         expected = compute_expected(n, p, k)
> >         output_values = list(map(int, captured_output.strip().split()))
> >         assert len(output_values) == k + 1
> >         for r in range(k + 1):
> >             assert output_values[r] == expected[r], f"Mismatch at r={r}: expected {expected[r]}, got {output_values[r]}"
> >
> > def compute_expected(n, p, k):
> >     """
> >     Computes expected results using brute-force for small n and p.
> >     """
> >     def matrix_rank(matrix):
> >         """Returns the rank of a square matrix over GF(2)."""
> >         rank = 0
> >         size = len(matrix)
> >         mat = [row[:] for row in matrix]
> >         for col in range(size):
> >             pivot = -1
> >             for row in range(rank, size):
> >                 if mat[row][col] == 1:
> >                     pivot = row
> >                     break
> >             if pivot == -1:
> >                 continue
> >             mat[rank], mat[pivot] = mat[pivot], mat[rank]
> >             for row in range(size):
> >                 if row != rank and mat[row][col] == 1:
> >                     for c in range(col, size):
> >                         mat[row][c] ^= mat[rank][c]
> >             rank += 1
> >         return rank
> >
> >     flat_size = n * n
> >     counts = [0] * (k + 1)
> >     for bits in product(range(p), repeat=flat_size):
> >         matrix = []
> >         idx = 0
> >         for i in range(n):
> >             row = []
> >             for j in range(n):
> >                 row.append(bits[idx])
> >                 idx += 1
> >             matrix.append(row)
> >         r = matrix_rank(matrix)
> >         if r <= k:
> >             counts[r] += 1
> >     return counts[:k+1]
> > ```

---

> > > ### Author Response · Authors · 2025-11-27
> > >
> > > **W3: Lack of related baselines**
> > >
> > > We thank the reviewer for the suggested related works. We will add a discussion with them to our revision. To clarify, we didn’t compare with these works for the following three reasons.
> > > 1. Different focus: These works focus on building **a pipeline with frozen LLMs** to test real-world repositories. LLMs are used as part of the pipeline, which they integrate with software engineering techniques like fuzzing to better test a system. In contrast, we focus on **improving the LLMs’ core reasoning ability** to generate test cases **through training**, and to study this problem, we evaluate on function-level testing as an abstraction of the problem.
> > > 2. Different goal: These works aim to improve the test coverage of the system. For that reason, some of them only generate test inputs to the system, without the associated test oracle, like assertions of expected outputs. Moreover, [5] focuses on regression testing, where they assume the system under test is correct, but our task is to distinguish whether a given code is correct or not, and we require the model to implement both the input generator and output verifier.
> > > 3. Different interaction pattern: The generation in these works is **iterative**: they use coverage or other execution feedback to repeatedly call the LLM to refine test inputs, but our method issues **a single LLM query** to generate test cases.
> > >
> > > Nevertheless, we have added **an additional baseline** similar to what the reviewer suggested in **W2**. Specifically, we compare with TrickCather [6], which uses the LLM to first generate multiple repaired programs of the program under test (PUT). It then generates an input generator, which is executed to produce test inputs. Finally, it collects test cases where the PUT produces different outputs from the repaired programs and uses the majority output of the repaired program as the expected output.
> > >
> > > The following table shows the performance of TrickCatcher when using Qwen3-4B as the base model (training-free), as well as the performance of harness generation when using Qwen3-4B (before training) and the model after RL training. We observe that TrickCatcher achieves performance comparable to our harness generation when both rely on a frozen base model. However, TrickCatcher incurs substantially higher computational cost: for each sample, it generates multiple repaired program variants (four in our implementation) and an input generator, resulting in four additional LLM queries per task than our method. In contrast, our method generates a complete harness in a single model call. The results show that our method is more efficient and achieves better performance.
> > > |  | LCB Unseen | |  | CF Unseen | |  |
> > > |---|---|---|---|---|---|---|
> > > |  | GI ↑ | ITR ↓ | TBR ↑ | GI ↑ | ITR ↓ | TBR ↑ |
> > > | TrickCatcher (frozen) | 25.9 | 42.9 | 8.1 | 31.1 | 46.3 | 5.2 |
> > > | Harness (frozen) | 28.9 | 44.6 | **8.3** | 29.2 | 50.1 | **6.1** |
> > > | Harness (RL) | 51.1 | 17.7 | **37.2** | 67.3 | 26.8 | **37.9** |
> > >
> > > **W4: The benefit of the proposed prompt for input generation is unclear, especially over existing techniques like random fuzzing**
> > >
> > > We want to clarify that the difference between our input generation and existing techniques like search-based generation or random fuzzing is that their generation is an iterative process that leverages execution feedback such as coverage analysis, whereas our input generator involves a single LLM query and requires the LLM to reason about the logic of the given code (e.g., analyzing where is the code wrong and designing inputs that target the bugs). Moreover, our focus is to improve the LLM’s ability to generate test cases through training, and we start with the current single query setting due to the complexity of RL training. We will leave the training of search-based generation or fuzzing, which involves multi-turn RL, to future work.
> > >
> > > [5] Pizzorno et al., CoverUp: Effective High Coverage Test Generation for Python.
> > >
> > > [6] Liu et al., LLM-Powered Test Case Generation for Detecting Bugs in Plausible Programs.

---

> > > > ### Author Response · Authors · 2025-11-27
> > > >
> > > > **W5: The evaluation can be enhanced by incorporating more languages and LLMs**
> > > >
> > > > Thanks for the suggestion. We are currently training a larger LLM (7B) and will update the results if it completes in time. Additionally, our experiments on Llama3.2-3B reported in the Appendix validate that the improvements of our method generalize to different base LLMs.
> > > >
> > > > Moreover, we have evaluated on **two additional versions** of the datasets, where the buggy programs are sampled from openai/gpt-oss-20b, which is from a completely different model family. The results in the following table demonstrate that our method still consistently outperforms the baseline method, indicating that the observed trends can generalize beyond the models seen during training. We will follow the suggestion to evaluate on more diverse programming languages.
> > > > |  |  | LCB |  |  | CF |  |
> > > > |---|---|---|---|---|---|---|
> > > > |  | GI ↑ | ITR ↓ | TBR ↑ | GI ↑ | ITR ↓ | TBR ↑ |
> > > > | I/O (RL) | 24.1 | 20.9 | 20.7 | 43.5 | 29.3 | 28.9 |
> > > > | Harness (RL) | 39.3 | 21.5 | **24.7** | 63.7 | 26.6 | **30.5** |
> > > >
> > > > **W6: The evaluation is limited. It is suggested to evaluate on real-world projects**
> > > >
> > > > Thanks for the suggestion. Please refer to our response in **W1** for why we use function-level testing as an abstraction to evaluate LLMs’ ability to generate test cases. Furthermore, we have evaluated on **an additional benchmark** beyond competitive programming. In particular, we evaluate on UnLeakedTestBench (ULT) [7], which targets functions for real-world coding problems. The following table shows the performance on ULT-Lite, where we report LCov@k and BCov@k, which measure the line coverage and branch coverage of the first k unique generated test cases. As can be observed in the table, our method consistently achieves higher line and branch coverage than the baseline, which suggests that the findings in this paper can be generalized beyond the competitive programming tasks.
> > > >
> > > > |  | Lcov@4 | Bcov@4 | Lcov@16 | Bcov@16 |
> > > > |---|---|---|---|---|
> > > > | Input/Output | 66.32 | 80.67 | 73.83 | 84.56 |
> > > > | Harness | **70.38** | **82.20** | **78.41** | **87.14** |
> > > >
> > > > [7] Huang et al., Benchmarking LLMs for Unit Test Generation from Real-World Functions.

---

### Official Review · Reviewer_Y4jL · 2025-10-29

**Soundness:** 2
**Presentation:** 2
**Contribution:** 2
**Rating:** 2
**Confidence:** 4

**Summary:**

The authors propose HarnessLLM, a framework that enables large language models to generate complete test harnesses, including both input generators and output validators, rather than simple input–output pairs. Through a two-stage training pipeline that combines supervised fine-tuning and reinforcement learning with verifiable rewards, the framework allows smaller models to effectively learn from larger teacher models and achieve performance comparable to or exceeding them in bug detection tasks.

**Strengths:**

1. The proposed framework advances beyond the conventional comparison between the expected output and the target program’s output. By incorporating invariant checking and brute-force reference implementations, it enables more fine-grained and reliable validation of program behavior.

2. The proposed framework demonstrates that a smaller model such as Qwen3-4B, when fine-tuned on outputs from the teacher model and further optimized with the proposed verifiable-reward-based RL, can even surpass the teacher’s performance. This result highlights the efficiency of the two-stage training pipeline and shows that the method can be effectively applied to smaller models without relying solely on large-scale LLMs, which enhances its practical value and scalability.

**Weaknesses:**

1. The experimental comparison remains narrow. Most results are derived from Qwen3-4B as the student model and Qwen3-32B as the sole teacher model, with only a supplementary experiment on LLaMA3.2-3B presented in the appendix. This limited setup makes it unclear whether the proposed approach generalizes to different model scales, architectures, or teacher–student combinations.

2. The experiment compares the proposed method only against UTGen and simple input–output testing. It would be helpful to include recent bug detection methods such as Fuzz4All [1] or TrickCatcher [2]. The limited baselines make it difficult to validate the claimed improvements and assess the method’s competitiveness.

3. Across Figures 5 and 7, and throughout much of the paper, the analysis remains largely quantitative. While the results present clear numerical trends, there is little causal or qualitative discussion explaining why such patterns occur. The absence of deeper interpretation limits a comprehensive understanding of the proposed method’s improvements and its behavior under different conditions. It would strengthen the paper to include qualitative analyses or case studies that illustrate how the proposed approach achieves its improvements or fails under specific scenarios.

4. The paper presents the validation stage as a central component of the test harness framework. The model-generated check_output() function is used to check invariants or compare results against brute-force or hardcoded outputs. However, there is no evaluation of how accurate or reliable these validation functions are. This limitation is critical because the generated verifiers are used in multiple stages. If these verifiers contain errors, the resulting noise can affect both training and evaluation. Without a quantitative or qualitative analysis of their reliability, the validity of the reported improvements remains uncertain.

[1] Chunqiu Steven Xia, Matteo Paltenghi, Jia Le Tian, Michael Pradel, and Lingming Zhang.  “Fuzz4All: Universal Fuzzing with Large Language Models.”, Proceedings of the 46th IEEE/ACM International Conference on Software Engineering, ICSE, 2024.
[2] Kaibo Liu, Zhenpeng Chen, Yiyang Liu, Jie M. Zhang, Mark Harman, Yudong Han, Yun Ma, Yihong Dong, Ge Li, and Gang Huang. “LLM-Powered Test Case Generation for Detecting Bugs in Plausible Programs”, In Proceedings of the 63rd Annual Meeting of the Association for Computational Linguistics, ACL 2025.

**Questions:**

1. The authors should clarify the reason for using Qwen3-32B as the sole teacher model for SFT data generation. It remains unclear whether this choice was made due to the model’s reasoning capability, accessibility, or compatibility with the student model. Providing an explanation or additional justification would help readers understand whether the results depend on this specific teacher configuration or can generalize to other large models.

2. It would be helpful if the authors reported the original performance of Qwen3-4B in Table 1 before applying SFT and RL training. Including this baseline would allow readers to clearly assess how much improvement is achieved through the proposed two-stage training pipeline.

3. Since both the teacher model and baselines are based on the Qwen family, the authors should clarify why the UNSEEN setting also uses buggy programs generated by Qwen3-14B. This design choice makes it unclear whether the UNSEEN experiments truly evaluate generalization beyond models within the same family.

---

> ### Author Response · Authors · 2025-11-26
>
> We thank the reviewer for the valuable comments and address each concern below.
>
> **W1 & Q1: Experimental comparison is narrow. It’s unclear whether the approach generalizes to different model scales, architectures, and teacher-student combinations**
>
> We want to clarify that we chose Qwen3-32B as the teacher model because it was the strongest reasoning model that we could practically deploy on our available hardware at the time of experimentation. Since the focus of this work is not on the teacher selection, we fix a single strong teacher and ensure a fair comparison between our harness-generation and the input–output baseline by (i) using the same teacher model and (ii) generating the same amount of SFT data.
>
> Additionally, our experiments show that the second-stage RL training brings larger improvements than the SFT training. To clarify the impact of each training stage, we now report the true bug rate (overall performance of finding a correct bug) of Qwen3-4B before any training, after SFT training, and after RL training. As can be seen, the overall average improvement from SFT to RL (20.2) is larger than that from base to SFT (16.8). Furthermore, for the base model and SFT model, the input-output baseline is better than harness generation on 4 and 3 out of 5 datasets, respectively, whereas for the RL model, our method outperforms the baseline on all 5 datasets. These results suggest that the **RL training is the key factor in our pipeline**.
> |  | MBPP+ | LCB Seen | CF Seen | LCB Unseen | CF Unseen | Avg |
> |---|---|---|---|---|---|---|
> | I/O (base) | 28.6 | **27.0** | **43.0** | **10.8** | **14.3** | **24.7** |
> | Harness (base) | **57.1** | 24.0 | 13.0 | 8.3 | 6.1 | 21.7 |
> | I/O (SFT) | 44.6 | **42.9** | **59.5** | **17.3** | 23.1 | 37.5 |
> | Harness (SFT) | **62.9** | 42.1 | 46.0 | 16.3 | **25.3** | **38.5** |
> | I/O (RL) | 72.7 | 65.1 | 72.2 | 33.3 | 32.4 | 55.1 |
> | Harness (RL) | **74.1**  | **69.9** | **74.4** | **37.2** | **37.9** | **58.7** |
>
> In addition, our experiments with a different student model, Llama3.2-3B (reported in the appendix), corroborate that the observed trends are not tied to a specific pair of teacher and student models. We are also training larger student models (7B) following the same pipeline and will update the results if they complete in time.
>
> **W2: The compared baselines are limited.**
>
> We follow the reviewer’s suggestions to compare with the additional baseline. Specifically, we compare with the TrickCatcher baseline on the two unseen versions of datasets. We do not include the other suggested Fuzz4All baseline because it is designed for system-level fuzzing instead of function-level test case generation. Additionally, it only generates test inputs instead of a complete pair of input and expected output behavior, which falls outside the scope of our task.
>
> The following table shows the performance of TrickCatcher when using Qwen3-4B as the base model (training-free), as well as the performance of harness generation when using Qwen3-4B (before training) and the model after RL training. We observe that TrickCatcher achieves performance comparable to our harness generation when both rely on a frozen base model. However, TrickCatcher incurs substantially higher computational cost: for each sample, it must generate multiple repaired program variants (four in our implementation) and an input generator, resulting in four additional LLM queries per task than our method. In contrast, our method generates a complete harness in a single model call. The results show that our method is **more efficient and achieves better performance**.
> |  | LCB Unseen | |  | CF Unseen | |  |
> |---|---|---|---|---|---|---|
> |  | GI ↑ | ITR ↓ | TBR ↑ | GI ↑ | ITR ↓ | TBR ↑ |
> | TrickCatcher (frozen) | 25.9 | 42.9 | 8.1 | 31.1 | 46.3 | 5.2 |
> | Harness (frozen) | 28.9 | 44.6 | **8.3** | 29.2 | 50.1 | **6.1** |
> | Harness (RL) | 51.1 | 17.7 | **37.2** | 67.3 | 26.8 | **37.9** |

---

> > ### Author Response · Authors · 2025-11-26
> >
> > **W3: Lack of qualitative analyses for the proposed method**
> >
> > The harness generation can outperform the input-output baseline because it enables **more diverse testing strategies**. Particularly, on the input side, it can directly hardcode test inputs, and it can also produce inputs through executable code, which makes it easier to generate richly structured and complicated inputs. On the output side, it can (1) hardcode expected output, (2) implement a correct function to produce expected outputs, or (3) check the expected properties or requirements of the outputs (please see the end of the response for examples of each strategy). More importantly, through RL training, the model learns to **adapt its testing strategies to the input problem**. For example, Figure 8 shows the distribution of each strategy in the trained model. When the coding problem is easy, the model more frequently generates a correct implementation to produce expected outputs. When the problem is difficult, the model looks for edge cases and hardcodes more expected outputs. Because of the larger space of testing strategies and the ability to adapt to a suitable strategy based on the given problem, our method can achieve better performance than the baseline, as indicated in the paper.
> >
> > **W4: The accuracy of the generated output checker is not evaluated**
> >
> > We would like to clarify that **our evaluation metrics and training rewards already capture the accuracy of the output checker**. Specifically, the true bug rate measures the percentage of test cases on which the ground truth program passes, but the buggy program fails. For a test case to identify a true bug, the ground truth program has to pass the test case, which means the output checker is correct for that specific input. In fact, the metric requires that both the input generator and output checker be correct.
> >
> > Nevertheless, we design **an additional metric to specifically evaluate the accuracy of the output checker**. In particular, we measure the output checker accuracy as # correct output / # correct input, where # correct output is the number of test cases where the output assertion passes for the ground truth program, and # correct input is the number of test cases where the input is valid (i.e., does not trigger any runtime error on the ground truth program). In other words, the output checker accuracy measures the fraction of correct output assertions among the valid test inputs (note that it is possible that the assertion does not trigger errors in the buggy program, but it is still correct because the ground truth program passes the assertion). The following table on the two unseen versions of datasets suggests that our method has a higher output checker accuracy than the baseline, which also validates our motivation that using executable code to verify output is more powerful than hardcoded expected outputs.
> > |  | LCB Unseen | CF Unseen |
> > |---|---|---|
> > | Input/Output (RL) | 87.9 | 71.1 |
> > | Harness (RL) | **90.8** | **86.4** |
> >
> > **Q2: Missing performance of Qwen3-4B before any training**
> >
> > Thanks for the suggestion. The results in our response to **W1 & Q1** show the base model’s performance without any training. As can be observed, the SFT and RL training bring significant improvements over the base model for both methods. Additionally, the SFT and RL training lead to larger improvements for our harness generation, which suggests that our proposed two-stage training pipeline unlocks the potential of harness generation. We will add this result to our revision.
> >
> > **Q3: Missing experiments where the buggy programs are generated by a non-Qwen model**
> >
> > We follow the reviewer’s suggestion to evaluate on **two additional unseen datasets**. Specifically, we use openai/gpt-oss-20b to generate candidate programs and filter for buggy programs using the same procedure described in the paper. This makes sure that the buggy programs are generated from a completely different model family. The results in the following table demonstrate that our method still consistently outperforms the baseline method. Combined with the Llama results in our appendix, they indicate that the **improvements of our method can generalize beyond the models seen during training**.
> > |  |  | LCB |  |  | CF |  |
> > |---|---|---|---|---|---|---|
> > |  | GI ↑ | ITR ↓ | TBR ↑ | GI ↑ | ITR ↓ | TBR ↑ |
> > | I/O (RL) | 24.1 | 20.9 | 20.7 | 43.5 | 29.3 | 28.9 |
> > | Harness (RL) | 39.3 | 21.5 | **24.7** | 63.7 | 26.6 | **30.5** |

---

> > > ### Author Response · Authors · 2025-11-26
> > >
> > > Below are examples of our model using different strategies for testing. We will add these examples to our revision.
> > >
> > > Here is an example where the model hardcodes input and output pairs:
> > > ```
> > > def generate_input_1() -> List[str]:
> > >     input_list = [
> > >         "3\n10 200 300\n4\n1 2 2 3\n",
> > >         "3\n1 2 3\n4\n1 2 2 3\n"
> > >     ]
> > >     return input_list
> > >
> > > def check_output(generated_input: str, captured_output: str) -> None:
> > >     captured = captured_output.strip()
> > >     if generated_input == "3\n10 200 300\n4\n1 2 2 3\n":
> > >         assert captured == "-1"
> > >     elif generated_input == "3\n1 2 3\n4\n1 2 2 3\n":
> > >         assert captured == "3"
> > > ```
> > >
> > > Here is an example where the model generates code to produce random inputs and checks the properties of the output:
> > > ```
> > > def generate_input_2() -> List[str]:
> > >     input_list = []
> > >     for _ in range(3):
> > >         n = random.randint(1, 100)
> > >         input_list.append(str(n))
> > >     return input_list
> > >
> > > def check_output(generated_input: str, captured_output: str) -> None:
> > >     """
> > >     Validate that every number from 1 to n appears exactly twice in the output.
> > >     """
> > >     n = int(generated_input.strip())
> > >     expected_length = 2 * n
> > >     output_list = list(map(int, captured_output.strip().split()))
> > >
> > >     # Ensure the output has the correct length
> > >     assert len(output_list) == expected_length, f"Output length mismatch: expected {expected_length}, got {len(output_list)}"
> > >
> > >     # Count occurrences of each number
> > >     freq = [0] * (n + 1)
> > >     for num in output_list:
> > >         assert 1 <= num <= n, f"Invalid number detected: {num} (out of bounds for n={n})"
> > >         freq[num] += 1
> > >
> > >     # Verify each number appears exactly twice
> > >     for i in range(1, n + 1):
> > >         assert freq[i] == 2, f"Number {i} appears {freq[i]} times instead of 2"
> > > ```
> > >
> > > Here is an example where the model implements a brute-force function to produce expected outputs:
> > > ```
> > > def check_output(generated_input: str, captured_output: str) -> None:
> > >     """
> > >     Validates the output for small cases via brute-force comparison.
> > >     """
> > >     parts = generated_input.strip().split()
> > >     n = int(parts[0])
> > >     p = int(parts[1])
> > >     k = int(parts[2])
> > >
> > >     # Brute-force for small n and p
> > >     if n <= 4 and p <= 7:
> > >         expected = compute_expected(n, p, k)
> > >         output_values = list(map(int, captured_output.strip().split()))
> > >         assert len(output_values) == k + 1
> > >         for r in range(k + 1):
> > >             assert output_values[r] == expected[r], f"Mismatch at r={r}: expected {expected[r]}, got {output_values[r]}"
> > >
> > > def compute_expected(n, p, k):
> > >     """
> > >     Computes expected results using brute-force for small n and p.
> > >     """
> > >     def matrix_rank(matrix):
> > >         """Returns the rank of a square matrix over GF(2)."""
> > >         rank = 0
> > >         size = len(matrix)
> > >         mat = [row[:] for row in matrix]
> > >         for col in range(size):
> > >             pivot = -1
> > >             for row in range(rank, size):
> > >                 if mat[row][col] == 1:
> > >                     pivot = row
> > >                     break
> > >             if pivot == -1:
> > >                 continue
> > >             mat[rank], mat[pivot] = mat[pivot], mat[rank]
> > >             for row in range(size):
> > >                 if row != rank and mat[row][col] == 1:
> > >                     for c in range(col, size):
> > >                         mat[row][c] ^= mat[rank][c]
> > >             rank += 1
> > >         return rank
> > >
> > >     flat_size = n * n
> > >     counts = [0] * (k + 1)
> > >     for bits in product(range(p), repeat=flat_size):
> > >         matrix = []
> > >         idx = 0
> > >         for i in range(n):
> > >             row = []
> > >             for j in range(n):
> > >                 row.append(bits[idx])
> > >                 idx += 1
> > >             matrix.append(row)
> > >         r = matrix_rank(matrix)
> > >         if r <= k:
> > >             counts[r] += 1
> > >     return counts[:k+1]
> > > ```

---

### Official Review · Reviewer_W71H · 2025-10-31

**Soundness:** 3
**Presentation:** 2
**Contribution:** 3
**Rating:** 4
**Confidence:** 4

**Summary:**

HarnessLLM addresses the limitations of automatic test generation through a two-stage training pipeline that teaches LLMs to write harness code, which is are programs that both synthesise test inputs and validate outputs via flexible checks such as invariants. The model is trained using supervised fine-tuning followed by reinforcement learning with verification rewards and a custom reward design. Experimental results show that HarnessLLM significantly improves bug detection and testing strategy diversity over traditional input–output testing, and further enhances code generation quality through test-time scaling

**Strengths:**

The paper works on test case generation, a very important issue for automatic code generation.

The proposed approach that uses an automatic validator to validate test outputs is interesting.

The trained model is demonstrated to work well for the benchmarks selected.

**Weaknesses:**

1. Potential bias in invariant learning and distribution analysis

The paper assumes that invariants can be effectively extracted or generated from real code, which might be true for competitive programs. But in practice, real-world programs often lack explicitly stated or easily derivable invariants. As a result, the reported distributional analysis of invariants could be biased toward competitive code or simplified code snippets, limiting the generalizability of the findings to realistic software systems.

2. Overfitting and data leakage concerns

The paper’s overfitting analysis examines whether the model overfits to specific models rather than to tasks. However, overfitting can also occur at the coding task level, especially if the training data or fine-tuning data contain test cases for identical problems to those used in evaluation. This creates a risk of data leakage.

3. Limitations of evaluation benchmarks

The benchmarks used for evaluation appear to be simplified and not fully representative of real-world programming challenges. They are designed for code generation, not test generation. Consequently, the claimed improvements might not translate to realistic software engineering scenarios. It is recommended that the authors evaluate their approach on benchmarks that are designed for test generation purposely, especially contamination-free benchmarks, such as UnLeakedTestBench (https://arxiv.org/pdf/2508.00408), which explicitly mitigates test case leakage and includes diverse, more challenging real-world coding problems.

**Questions:**

What are the potential risks of data leakage in the evaluation process?

How does the model perform on test generation benchmarks?

---

> ### Author Response · Authors · 2025-11-26
>
> We thank the reviewer for the valuable comments and address each concern below.
>
> **W1 & W3: Bias and limitations of evaluation benchmarks**
>
> In line with the reviewer’s suggestion, we additionally evaluate our method and baselines on the UnLeakedTestBench (ULT) benchmark, which is particularly designed for test generation and covers challenging real-world coding problems. The results in the following table show that **our method consistently outperforms the RL-trained input-output baseline on the suggested benchmark**.
>
> Specifically, for efficiency, we evaluate on the officially released lite subset of ULT. Additionally, since the evaluation code is not available, we only report the LCov@k and BCov@k, which measure the line coverage and branch coverage of the first k generated test cases. For fair comparison, we measure the coverage under the same number of unique test cases for both methods. As can be observed in the table, our method consistently achieves higher line and branch coverage than the baseline. The results suggest that the findings in this paper can be generalized beyond the competitive programming tasks. We will include these additional results in our revision.
> |  | Lcov@4 | Bcov@4 | Lcov@16 | Bcov@16 |
> |---|---|---|---|---|
> | Input/Output | 66.32 | 80.67 | 73.83 | 84.56 |
> | Harness | **70.38** | **82.20** | **78.41** | **87.14** |
>
> **W2: Data leakage concerns**
>
> We would like to clarify that **all the training data in our paper has been decontaminated against the evaluation data** (line 241). Particularly, we remove all samples in the training set where the description of the coding problem is similar to that of any evaluation sample. We follow the community standard [1] to judge if two texts are similar or not. This procedure eliminates duplicates and near-duplicates of evaluation samples from the training data.
>
> [1] Yang et al., Qwen2.5-Math Technical Report: Toward Mathematical Expert Model via Self-Improvement.

---

### Note · Authors · 2026-01-05

I have read and agree with the venue's withdrawal policy on behalf of myself and my co-authors.